# New immunomodulatory treatment protocol for canine leishmaniosis reduces parasitemia and proteinuria

**Guadalupe Miró**[1]\*, **Sergi Segarra**[2], **José Joaquín Cerón**[3], **Lluís Ferrer**[4], **Laia Solano-Gallego**[4], **Laia Montell**[2], **Ester Costa**[2], **Joan Teichenne**[5], **Roger Mariné-Casadó**[5], **GALILEI trial Group**[¶], **Xavier Roura**[6]

**1** Animal Health Department, Veterinary School, Universidad Complutense de Madrid, Madrid Spain, **2** R&D Bioiberica S.A.U., Esplugues de Llobregat, Spain, **3** Interlab-UMU, Campus de Excelencia "Mare Nostrum", University of Murcia, Campus Espinardo, Murcia, Spain, **4** Departament de Medicina i Cirurgia Animals, Facultat de Veterinària, Universitat Autònoma de Barcelona, Bellaterra, Spain, **5** Eurecat, Technology Centre of Catalonia, Technological Unit of Nutrition and Health, Reus, Spain, **6** Hospital Clínic Veterinari, Universitat Autònoma de Barcelona, Bellaterra, Spain

¶ Membership of the GALILEI trial Group is provided in the Acknowledgments.
\* gmiro@ucm.es

**Data Availability Statement:** All data are in the manuscript and/or supporting information files.

**Funding:** The author(s) received no specific funding for this work.

## Abstract

The current standard treatment for canine leishmaniosis (CanL), N-methylglucamine antimoniate (MGA) given with allopurinol, is not fully effective and may cause adverse effects and drug resistance. *In vitro* and *in vivo* studies have shown that nucleotides, administered alone or with AHCC, offer benefits in the treatment of CanL. This study examines the effects of a new immunomodulatory treatment protocol in which dietary nucleotides and AHCC are added to the recommended standard treatment. Out of 160 sick dogs with naturally occurring clinical leishmaniosis recruited, 97 were randomized to a supplement (n = 47) or control (n = 50) group. All dogs received an initial 28-day course of MGA and 365-day course of allopurinol. From day 0 to day 730, dogs in the control group additionally received a placebo, while dogs in the supplement group received Impromune (Bioiberica S.A.U., Esplugues de Llobregat, Spain), an oral supplement providing 32 mg/kg nucleotides and 17 mg/kg AHCC daily. After 2 years, five dogs had relapsed in the supplement group (18.5%) while seven did so in the control group (22.6%). Over time, animals in both groups showed significant improvements in body weight, LeishVet clinical stage, clinical score, and anti-*Leishmania* antibodies. Adding the supplement to the standard protocol resulted in further significant improvements, namely in reducing the parasite load and urinary protein/creatinine ratio, improving IRIS stage, lowering serum creatinine levels on day 30, deceasing urine turbidity on day 365, and improving weight gain on day 545. The daily intake of the supplement over two years proved safe and well tolerated. Our study confirms the efficacy of the recommended standard treatment for CanL, but also reveals that by adding Impromune additional benefits are obtained, especially reduced parasitemia and improved renal function.

**Competing interests:** I have read the journal's policy and the authors of this manuscript have the following competing interests: SS, LM and EC were employees of Bioiberica S.A.U.

## Author summary

The leishmaniases are a group of neglected tropical diseases caused by protozoa that affect humans and other animals, whose world distribution areas are currently expanding due to climate change and globalization, among other factors. As dogs are the main reservoir of infection by the parasite, controlling the disease in this species should help minimize infection spread and reduce the prevalence of leishmaniasis in humans. Unfortunately, current treatment for sick dogs is not fully effective and there are reports of associated side effects and drug resistance. The results of this clinical trial in sick dogs indicate that adding an oral supplement (Impromune, Bioiberica S.A.U., Esplugues de Llobregat, Spain) containing nucleotides and AHCC to the standard treatment regimen for canine leishmaniosis serves to reduce the parasite load and improve kidney function. Accordingly, we would recommend this new immunomodulatory treatment protocol as a tool to optimize the global management of leishmaniasis in a One Health approach.

## Introduction

The leishmaniases are a group of complex diseases that affect different hosts, including humans, companion animals, livestock, and wildlife in many world regions. With the current rise in global temperatures due to climate change and increased movement of people and animals, this is now an emerging zoonosis expanding to non-endemic countries, and a public health concern for the years to come. In addition, the leishmaniases have been included in the list of neglected diseases [1–9]. Canine leishmaniosis (CanL) caused by *Leishmania infantum* is a major global zoonosis that can be fatal to dogs and humans. The prevalence of CanL in endemic areas can be as high as 30% to 50% in dogs, which are considered the main reservoir of infection [1,10,11].

Today, PCR-based molecular assays are among the main laboratory methods used for the diagnosis of infection and follow-up of patients with CanL [12]. Quantifying the *Leishmania* parasite load provides very relevant information on a particular case and is also important from a public health standpoint in relation to infection transmission.

Chemotherapy is a key strategy for the clinical management of CanL, and subcutaneous N-methylglucamine antimoniate (MGA) given for 4 to 6 weeks combined with oral allopurinol for at least 6 months is so far the most effective treatment. This combination of drugs with leishmanicidal (MGA) and leishmaniostatic (allopurinol) properties is the standard treatment recommended in the guidelines for the practical management of CanL issued by the LeishVet group [13] and Canine Leishmaniasis Working Group [14], although other options have also been shown effective, such as the combination treatment miltefosine and allopurinol [15–19]. However, while these rather expensive treatments may achieve clinical cure in a high percentage of treated dogs, parasitological cure is not observed in sick dogs. In addition, the most commonly used therapies can lead to side effects such as xanthine urolithiasis, renal mineralization, nephrolithiasis due to allopurinol [20–22], azotemia and/or proteinuria along with pain and inflammation at the MGA injection site [13,23,24], and dysorexia, vomiting, and mild self-limiting diarrhea due to miltefosine [13,16]. Further, treatment failure is quite common, and relapses are frequent. Drug resistance has been also reported [25–33], although there is controversy over this matter and the link between treatment failure and drug resistance is still unclear [34].

Given the importance of mounting an appropriate immune response to avoid relapse in CanL, a treatment approach including the use of immunomodulators has been suggested as a

future strategy [35–38]. Modulating the immune response seems an adequate way of clinically managing these dogs, given that the immune system plays a key role in disease progression [39–42]. In line with this rationale, some studies have explored the potential benefits of nucleotides and AHCC as treatment for CanL [43–46]. Nucleotides are low molecular weight bioactive compounds which are naturally found in all foods of animal and vegetable origin as free nucleotides and nucleic acids. Under normal conditions, *de novo* endogenous synthesis is enough to cover the nucleotide needs of animals including humans. A dietary nucleotide supply, however, becomes essential when the body is not able to produce enough to meet demands [47–49]. Products obtained from brewer's yeast are rich in nucleotides. Hence, yeast extracts could be used as immunomodulators [50,51]. AHCC, a standardized extract of cultured *Lentinula edodes* mycelia, especially rich in α-glucans, has been found to offer beneficial health effects such as modulation of the immune response towards Th1, antioxidant and anticancer activity, and the prevention of infections [52–55].

The use of yeast-derived nucleotides, alone or in combination with AHCC, has emerged as effective against *L. infantum* infection *in vitro* [44] and CanL *in vivo* [45,46], as well as a coadjuvant treatment in cats with leishmaniosis [56–58]. *In vitro*, these compounds enhance the effective Th1 immune response against *L. infantum* mainly by increasing the release of IFN-γ and TNF-α in naïve- and *L. infantum*-infected macrophage/lymphocyte cocultures [44]. *In vivo*, two clinical trials support the clinical efficacy and safety of such a combination. More specifically, in a multicenter, open-label, positively-controlled randomized clinical trial in dogs with clinical leishmaniosis receiving an initial 28-day course of MGA, the administration of nucleotides plus AHCC for six months gave rise to significantly better clinical efficacy than allopurinol, without promoting xanthinuria [45]. In another multicenter, double-blinded, placebo-controlled randomized trial conducted in clinically healthy *Leishmania*-infected dogs, the effects of nucleotides plus AHCC were compared to those of placebo. After one year, a significantly lower proportion of dogs showed disease progression in the supplement group. Moreover, anti-*Leishmania* antibodies were reduced only in the group of dogs treated with the supplement, and the mean clinical score of disease severity was significantly lower in the supplement group after 180 days [46]. Besides these trials, other studies have also shown that this combination can provide benefits for the treatment of feline leishmaniosis [57,58].

The aim of this study was to assess the efficacy and safety of a new immunomodulatory treatment protocol for CanL consisting of adding nucleotides and AHCC to the recommended standard MGA-allopurinol treatment.

## Methods

### Ethics statement

Ethics approval for the study protocol was obtained from the Animal Experimentation Ethics Committee (CEEA) of the University of Murcia, Murcia, Spain (REGA ES300305440012; CEEA code 480/2018), and the Ethics Committee of the Faculty of Veterinary Medicine–University of Lisbon (CEBEA; code 007/2019). All dog owners gave their written informed consent.

This was a multicenter, randomized, double-blind, placebo-controlled clinical trial. Client-owned dogs of any age, breed, or gender, with naturally occurring leishmaniosis were recruited from 13 veterinary practices in regions of Spain and Portugal, where CanL is endemic.

Dogs with clinical signs and/or clinicopathological abnormalities associated with clinical leishmaniosis and classified as LeishVet stage II or III [59] were included in the study if they showed either a high antibody titer (> 500 *Leishmania* units of fluorometry (LUF)) in time resolved-immunofluorometric assay (TR-IFMA) quantitative serology [60,61], or a medium

TR-IFMA titer (50–500 LUF) together with a positive peripheral blood quantitative polymerase chain reaction (qPCR) result. Dogs were not included if they had been vaccinated against CanL, had received treatment with allopurinol, MGA, miltefosine, domperidone, cyclosporine or glucocorticoids, or other immunomodulating and/or immunosuppressive drugs (such as oclacitinib, azathioprine, or mofetil mycophenolate) four months before entering the study, or if they were receiving any kind of special diet or dietary supplements to improve their immune response. Neither were pregnant or lactating females included. Dogs were excluded if they had chronic kidney disease (International Renal Interest Society (IRIS) stage $\geq 3$) or were diagnosed with any type of neoplasia. Additionally, dogs could be withdrawn from the study at any time if they showed intolerance to the interventions used or if requested by the pet owner.

Dogs that met the inclusion and exclusion criteria were randomized using a computer-generated schedule to one of two study groups: supplement group (standard treatment plus an oral supplement) and control group (standard treatment plus placebo). Treatment started immediately after enrollment. All dogs received an initial 28-day course of MGA (50 mg/kg SC BID or 100 mg/kg SC SID) and a 365-day schedule of allopurinol (10 mg/kg orally BID). From Day 0 till Day 730, dogs in the control group additionally received an oral placebo, while dogs in the supplement group were administered an oral supplement (Impromune, Bioiberica S.A.U., Esplugues de Llobregat, Spain) containing a nucleotide-rich yeast extract providing a daily amount of 32 mg/kg BW of nucleotides (Nucleoforce, Bioiberica S.A.U., Esplugues de Llobregat, Spain) and 17 mg/kg BW of AHCC (Immunactive, Amino Up Co. Ltd., Sapporo, Japan). Dogs in the placebo group administered inert microcrystalline cellulose oral tablets which shared the same physical appearance as the supplement tablets such that they received no antileishmanial treatment between days 365 and 730. Throughout the study, all dogs were fed a regular diet, although different trademarks and formulations were allowed. A prescription diet for leishmaniosis was not allowed.

Several efficacy and safety parameters were assessed, including clinical LeishVet staging [59], total clinical score [45], IRIS staging [62], clinical relapse rate, parasite load measured through blood qPCR, complete blood count (CBC), serum biochemistry, serum protein electrophoresis, acute phase proteins (C-reactive protein, ferritin, paraoxonase-1, haptoglobin and albumin) and TR-IFMA *Leishmania* quantitative serology. Urinalysis, including urinary protein/creatinine ratio (UP/C), urine specific gravity, and urinary sediment analysis, was also performed. Clinical relapses were defined as significant clinical worsening and/or laboratory abnormalities [16], which in normal clinical practice would require additional treatment against *Leishmania*, along with a marked increase in antibody levels. The final diagnosis of relapse was made by the principal investigator (G.M.) based on the details provided by the recruiting center and the practitioner responsible, details from clinical records and laboratory results. Once the infection relapse was pronounced, the dog was withdrawn from the study. During the follow-up period, any adverse effects that could be related to the compounds administered, such as gastrointestinal problems or urinary abnormalities, were recorded.

Results were compared over time and between groups at baseline, and after 30, 180, 365, 545 and 730 days (Fig 1). Between on-site visits, the investigators at each veterinary practice contacted the dog owners by phone on days 90, 270, 455 and 635. These interviews allowed the practitioners to make sure adequate progress was being made and to decide whether an extra on-site visit was needed to decide if there was a need for additional treatment before the next visit, or to record a relapse and accordingly proceed with the decision making.

Baseline differences were analyzed using Student's t-test for quantitative variables and Fisher's exact test for categorical variables. Treatment effects were compared by analysis of covariance (ANCOVA) using baseline values as co-variables for quantitative variables, and Fisher's exact test for categorical variables. Changes over time within each group were analyzed by

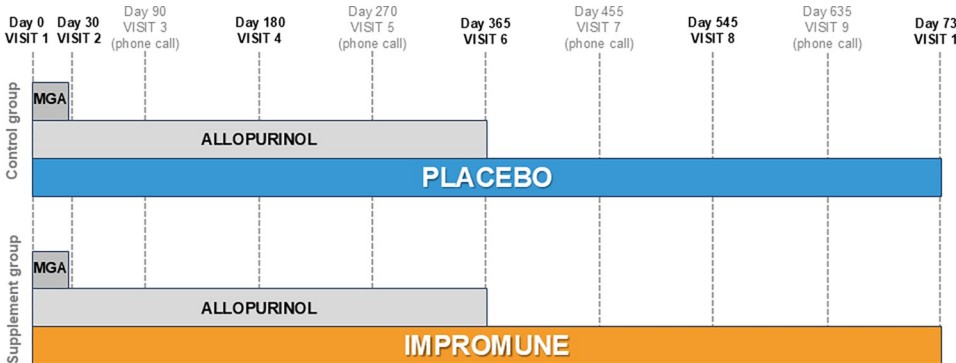

**Fig 1. Study design.** Interventions, follow-up visits, and phone calls over the course of the study in each group. MGA: N-methylglucamine antimoniate.

repeated-measure analysis of variance (rmANOVA) for quantitative variables, and McNemar's test for categorical variables. For statistical analysis of the data, only dogs with available data on the main efficacy variable (disease relapse) and for which there had been no major deviations of the protocol were included (per protocol population; PP population). Accordingly, statistical analyses only considered dogs completing the study (dogs with data recorded in the last on-site visit 730 days after the study start) and dogs not completing the study experiencing disease relapse.

## Results

Out of 160 dogs assessed for eligibility, 97 were randomized to the two treatment arms: standard treatment plus supplement (n = 47) and standard treatment plus placebo (n = 50). At baseline, no significant differences between study groups were recorded in demographic (sex, age, and breed) or other characteristics (body temperature and weight). Neither were there differences (control *vs* supplement group, respectively) in TR-IFMA serology (1009±916 *vs* 1265 ±1106 LUF; p = 0.344), qPCR (14.46±22.67 *vs* 16.19±24.28; p = 0.788), frequency of positive qPCR (19 (65.5%) *vs* 19 (76%) dogs; p = 0.552), LeishVet clinical stage (2.19±0.40 *vs* 2.37±0.49; p = 0.144) or total clinical score (9.00±5.58 *vs* 10.11±4.62; p = 0.416). Initially, there was a significantly higher (p = 0.047) proportion of non-proteinuric dogs (IRIS stage 1) observed in the control group (9 dogs; 29%) than supplement group (2 dogs; 7.4%).

Our final analysis included data from 58 dogs (supplement group n = 27; placebo group n = 31) (Fig 2): 29 males (16 in the control group, 13 in the supplement group) and 29 females (13 in the control group, 16 in the supplement group) of similar age (mean±SD: control group 5.63±3.35 years, supplement group 5.39±2.97 years). Participants comprised several breeds: crossbreed (n = 24), Teckel (n = 4), Cocker spaniel (n = 3), Pitbull (n = 3), Doberman (n = 3), German shepherd (n = 2), American Staffordshire (n = 2), Labrador retriever (n = 2), Yorkshire terrier (n = 2), Boxer (n = 1), Andalusian Bodeguero (n = 1), English setter (n = 1), Mastiff (n = 1), Belgian shepherd Malinois (n = 1), Schnauzer (n = 1), Spanish greyhound (n = 1), Podenco (n = 1), Czechoslovakian wolfdog (n = 1), Belgian shepherd (n = 1), Border collie (n = 1), Golden retriever (n = 1) and Laekenois (n = 1).

After the 2-year follow-up period, 5 dogs in the supplement group and 7 in the control group had suffered disease relapses (18.5% *vs* 22.6%; p> 0.05) (Fig 3).

The numbers of dogs not completing the study were 19 in the control group and 20 in the supplement group (p = 0.681). Reasons for withdrawal were mainly protocol violation or decision made by the pet owner. During the study, four dogs died in each study group for reasons

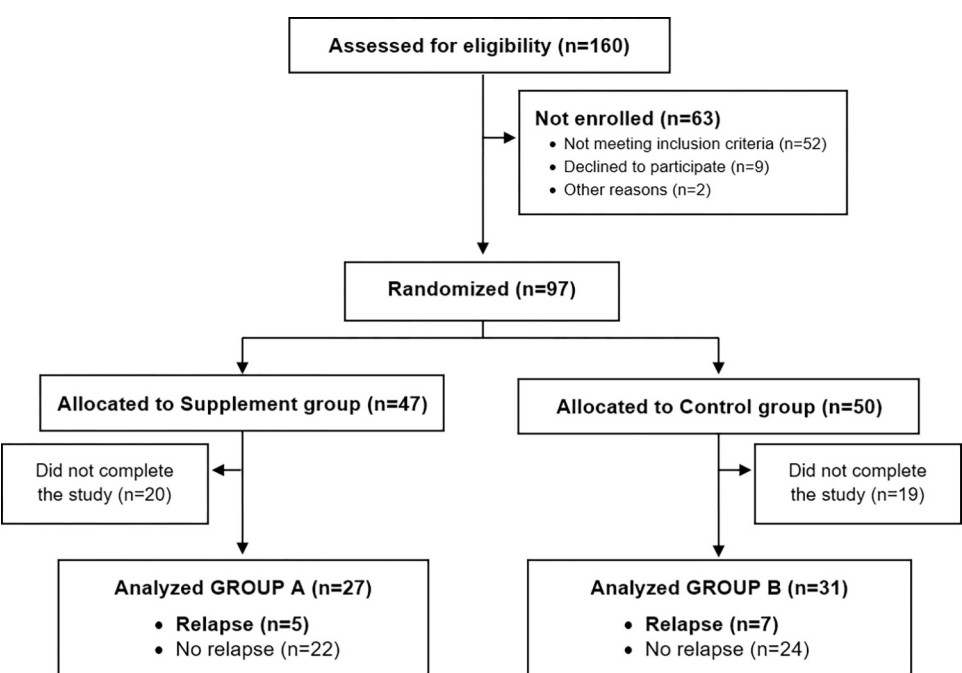

**Fig 2. Flow diagram.** Eligibility assessment, randomization to each treatment arm, and relapses produced.

not directly related to the therapy or disease. Main reasons were road traffic accident, and death or euthanasia due to diseases or other causes not directly related to CanL.

Significant improvements ($p < 0.05$) were observed over time in both groups in LeishVet clinical staging, total clinical score, and TR-IFMA anti-*Leishmania* serology (Fig 4). No significant differences were observed in these variables between groups at any time point.

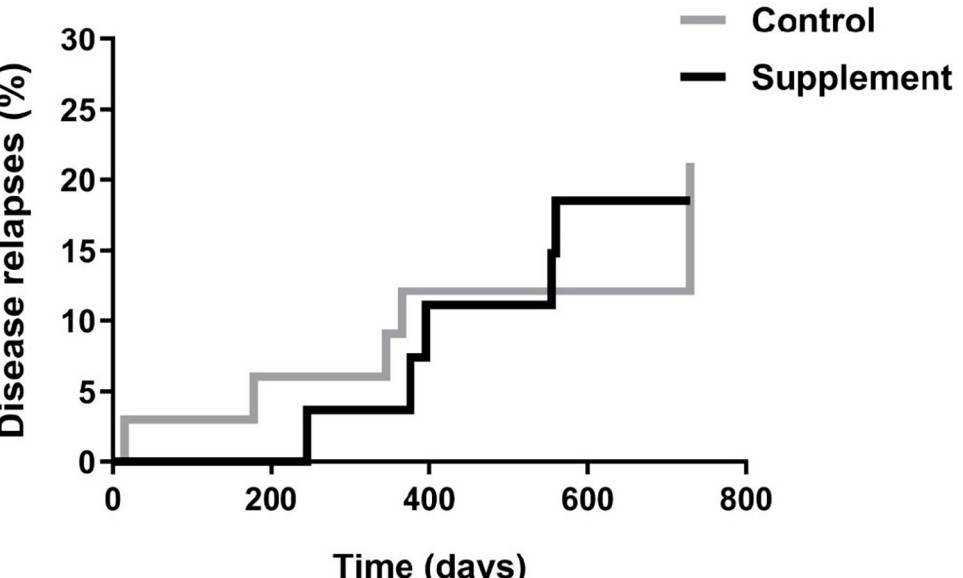

**Fig 3. Disease relapses.** Percentages of dogs experiencing relapses during the study period in each treatment group.

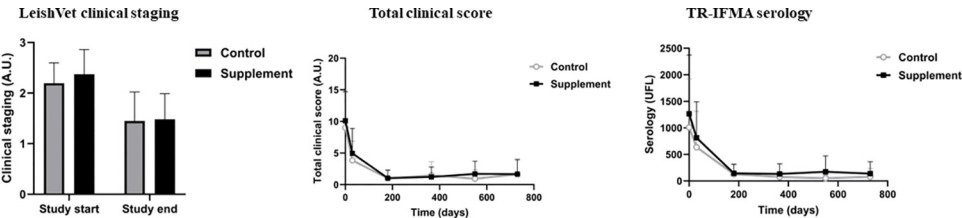

**Fig 4. LeishVet clinical staging, total clinical score and TR-IFMA anti-*Leishmania* serology.** Changes in the variables produced over time in each study group.

Although no significant differences were observed between groups in parasite load (blood qPCR), only the supplement group showed a significant decrease over time (p = 0.004), while the control group showed no significant change over time (Fig 5).

Over time, a significant decrease in UP/C (p = 0.008) was observed only within the supplement group, while no significant differences over time were observed in the control group. Also, compared to the control group, dogs in the supplement group showed significantly lower serum creatinine concentrations on day 30 (0.83±0.18 *vs* 0.92±0.23; p = 0.020) (Fig 6).

The supplement group showed significant improvement in IRIS staging over time, which did not occur in the control group. More specifically, a significant increase was recorded in the percentage of dogs classified as IRIS stage 1 non-proteinuric (1NP) from 0 to 730 days (p = 0.006) only in the supplement group (Fig 7).

Mean body weight on day 545 was significantly higher in the supplement group than control group (22.97±11.50 *vs* 19.07±10.98 kg; p = 0.016). Also in the supplement group, urine turbidity on day 365 was significantly lower (transparent urine frequency 63% vs 32.3%; p = 0.034) and urine pH was significantly higher at 30 days (7.30±0.97 *vs* 6.23±0.89; p = 0.001) and at 545 days (7.21±1.36 *vs* 6.48±1.07; p = 0.015).

No other significant changes over time or differences between groups were observed for the rest of the variables. Intake of the test products over two years proved safe and was well tolerated by all treated dogs. No significant differences emerged between groups in terms of frequency of adverse effects.

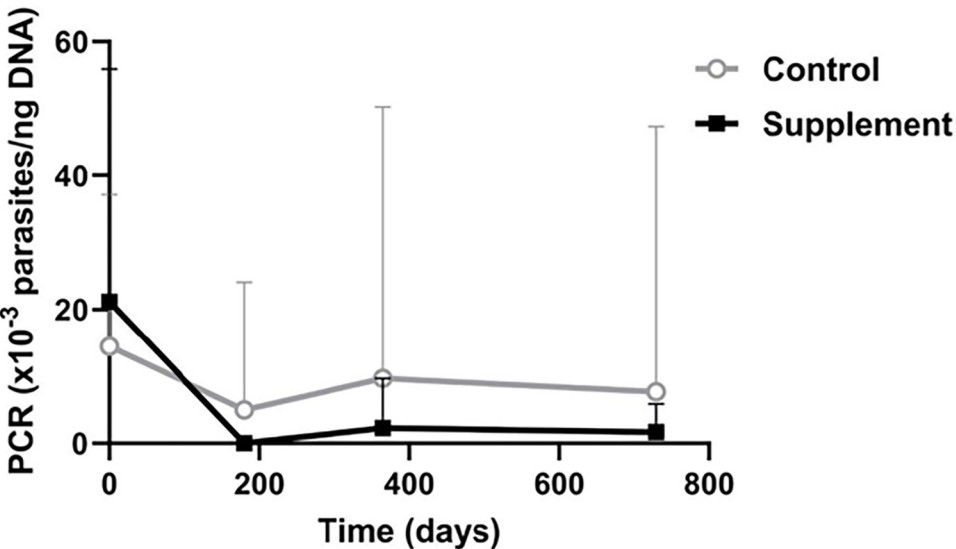

**Fig 5. Blood qPCR.** Changes in parasite load produced over the study course in each group.

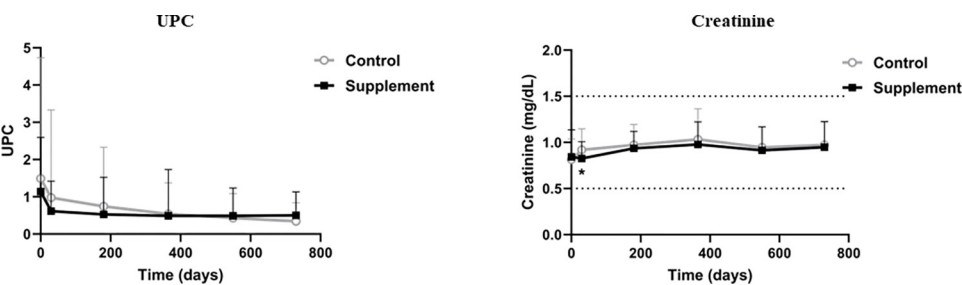

**Fig 6. UP/C and creatinine.** Changes in urinary protein/creatinine ratio (UP/C) and serum creatinine concentrations produced during the study in each group. *, p< 0.05 ANCOVA using baseline values as covariables.

## Discussion

Leishmaniases are among the most neglected and underreported diseases that can spread between humans and other animals. This means CanL is not only a concern for veterinarians

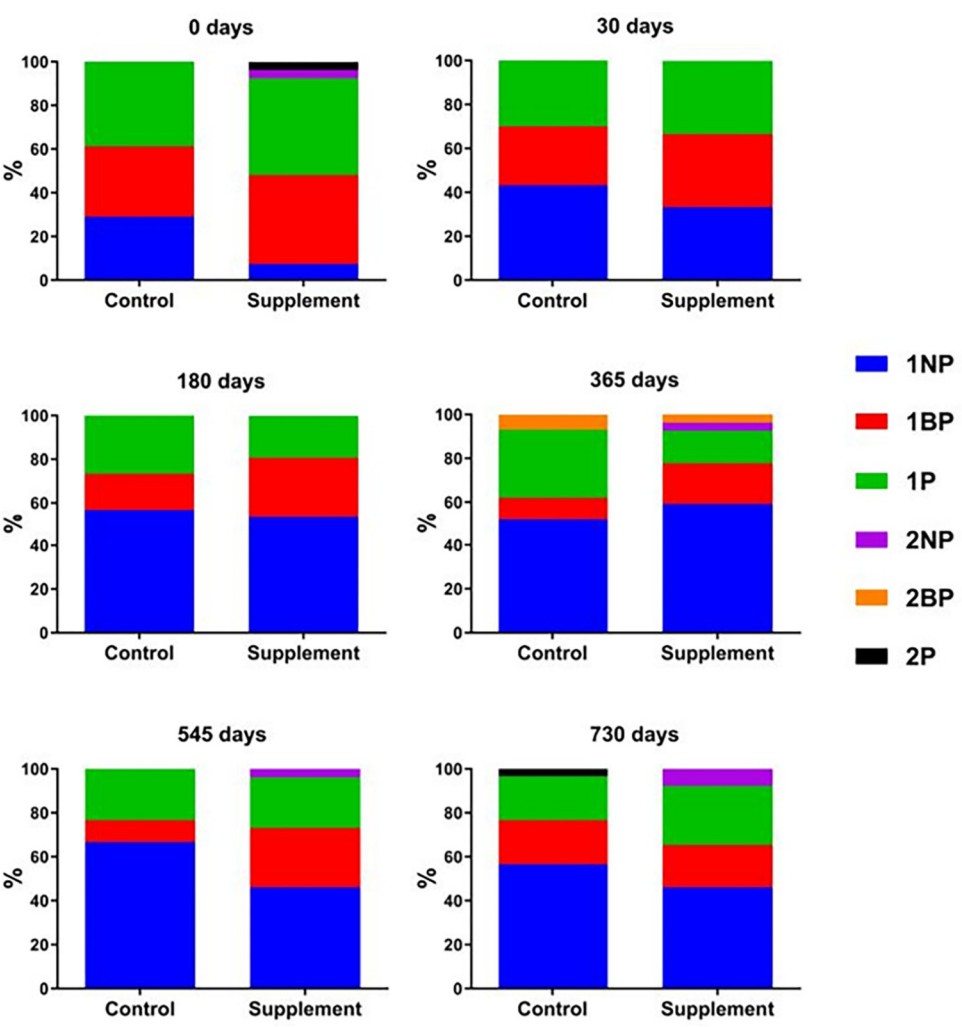

**Fig 7. IRIS staging.** IRIS stage at the established follow-up times in each study group expressed as percentages of dogs at each stage. 1NP: stage 1 non-proteinuric; 1BP: stage 1 borderline proteinuric; 1P: stage 1 proteinuric; 2NP: stage 2 non-proteinuric; 2BP: stage 2 borderline proteinuric; 2P: stage 2 proteinuric.

but is also a significant public health concern. Hence, a One Health approach is needed to globally control leishmaniasis including adequate management of the infection in dogs to reduce its prevalence in these animals and consequently also in humans and other animals [1,63–65].

For this purpose, the combination of MGA plus allopurinol has been the mainstay treatment for many years and is still the recommended standard treatment for CanL. In our study, we confirmed its efficacy but despite much research into this treatment [15,18,66], to the authors' knowledge, up until now it had not yet been investigated whether dogs with leishmaniosis that had received an initial 28-day course of MGA and one year administration of allopurinol could remain stable for another year after stopping such treatment protocol.

While the various drugs available against CanL do effectively serve to improve clinical status and reduce the parasite burden, no treatment regimen has so far been able to completely eliminate the infection or prevent its relapse [67,68]. Avoiding disease relapses is one of the key goals of the clinical management of CanL. Relapses usually occur after a variable period of time after treatment (from several months to several years) and will especially affect dogs in which the duration of MGA treatment is shorter than four weeks [14,69]. In addition, the clinical response to treatment may vary, depending on the initial clinical stage and the individual characteristics of each dog [13,70]. Veterinary practitioners thus try to keep their patients in clinical remission for as long as possible with the minimum use of drugs to avoid adverse effects and drug resistance. This situation determines a need for new therapeutic options to improve the clinical management of this disease. Among the new treatments, active agents with immunomodulatory properties are receiving much attention, given the important role played by the immune response in the pathophysiology of CanL [37,38,71,72].

In this study, the frequency of disease relapses was surprisingly low overall, which further supports the efficacy of the standard recommended treatment. There were only 22.6% cases of relapses in our control group, making it difficult to detect an additional beneficial impact of the supplement. In effect, relapse was produced in a lower, albeit not significant, percentage of dogs (18.5%) additionally given Impromune.

Although current CanL therapies are not able to eliminate the parasite, several studies have shown that parasite load is much reduced, as measured through qPCR [13,15,16,45,72]. This reduction in parasite load occurs in parallel with clinical improvement, and thus seems to help the dog's immune response control the infection and disease. It is therefore the main target of current therapy [10,12,73]. The significant decrease in qPCR values observed here over time in the long term only with the supplement indicates an additional benefit over the recommended standard protocol given the impact of the supplemented treatment on one of the hallmarks of CanL. We thus propose that nucleotides plus AHCC could produce a leishmaniostatic effect by keeping parasite levels low even in the absence of allopurinol treatment. It should also be considered that qPCR analyses were performed on blood samples and not bone marrow aspirates. Indeed, this quantification method has the advantage that blood is an easier to obtain, less invasive, and more stable biospecimen [74–76].

The deposition of circulating immune complexes in the glomeruli results in an immune-mediated glomerulonephritis which can provoke chronic kidney disease (CKD). This is one of the main complications of CanL. Starting with proteinuria, this complication may progress to azotemia and kidney failure and is the most common cause of losses to CanL [6,77–80]. Our results show a positive impact of the new immunomodulatory treatment protocol on kidney function, as indicated by improvements in serum creatinine levels, UP/C and IRIS staging. This suggests that Impromune added to the current standard CanL treatment could be useful for the management of sick dogs particularly those with impaired kidney function at the time of diagnosis. A possible explanation might be that the oral intake of nucleotides, AHCC, or

their combination, somehow improves the glomerulopathy from which CanL patients suffer over the course of the disease. High levels of serum creatinine are considered a negative prognostic factor in CanL [79] and this parameter was positively affected by this new protocol. Similar benefits for kidney function and proteinuria have been described for other therapies, including domperidone and miltefosine given with allopurinol [72,81,82]. Urinary pH was also significantly improved by the new treatment protocol, as a more alkaline urine was found in dogs receiving the supplement after 30 and 545 days. Promoting an alkaline urine can help minimize the risk of xanthinuria by decreasing the solubility of some lithogenic substances [83,84].

We observed no treatment effects on LeishVet staging, total clinical score or quantitative serology. This could probably be explained mainly by the efficacy of the standard treatment, which determined that both groups showed significant improvements in these key variables over time. It is worthwhile mentioning though that the supplement group happened to feature higher baseline mean LeishVet staging values, total clinical scores, and quantitative serology, although there were no significant differences between the groups at that time point.

Our findings point to possible benefits of adding Impromune to the standard CanL treatment protocol. This supplement could enhance the effective immune response against CanL and hence increase its efficacy. Such enhancement is probably achieved by the intensification of the Th1 cellular immune response mediated by nucleotides and AHCC, as described previously [44].

Based on our observations, this nutritional supplement could also diminish the side effects of the standard CanL treatment protocol by reducing the need for several MGA cycles, lowering the risk of inducing resistance in *L. infantum* strains. Further, given that MGA treatment requires daily injections meaning a high cost for the dog owner, this would translate to an easier more convenient treatment protocol associated with improved treatment compliance. In addition, nucleotides plus AHCC could also be used in apparently healthy dogs living in endemic areas. Some of these dogs might benefit from a potential disease prevention effect [43,46]. Others might actually be sick dogs with laboratory abnormalities which appear healthy on physical examination [73] which would also benefit from treatment. Our study also supports the safety of this supplement which, after daily administration for two years, was well tolerated and no adverse effects were reported in treated dogs.

Having a wide range of tools to manage CanL, especially considering the usefulness of the new immunomodulatory treatment protocol described herein, should help with the global control of this disease following a One Health approach. This takes on special relevance considering the ongoing geographical expansion of leishmaniosis [1,4,9,85,86].

Our study has some limitations. First, we had missing data for the period between days 30 and 180. However, given the trends observed in each group, this could be seen as a minor limitation. In addition, concomitant diseases were not evaluated during the study. This could have had a mild impact on our results given that associations have been reported between clinical CanL and other vector-borne infections [87]. Lastly, the sample size could have been larger, but screening failures and study drop-outs were higher than expected. It should be highlighted that clinical trials in pet dogs with naturally occurring CanL are scarce mainly due their complexity and the difficulty in recruiting a significant number of study subjects. Nonetheless, for a study of this type and with such a long follow-up period, it should be appreciated that the sample size was still quite large.

## Conclusion

This study confirms the efficacy of the current recommended treatment for CanL. The combination of MGA and allopurinol in our control group led to significant improvements in clinical and laboratory parameters after two years of follow-up, and to a low rate of disease relapse even during a second year without treatment. Our results also indicate that the new immunomodulatory treatment protocol incorporating nucleotides and AHCC offers additional benefits over the standard treatment. These mainly include significantly reduced parasitemia and proteinuria with no negative impact on the clinical status of treated dogs. The oral administration of Impromune daily for two years was shown to be safe. Based on its observed efficacy, it might be useful as part of the multimodal therapeutic strategy for the clinical management of CanL following a One Health approach. It could also contribute to reducing the overuse of antileishmanial drugs, with the consequences of preventing future resistances and diminishing the frequency of adverse effects of these drugs.

## Supporting information

**S1 Data. Raw data related signalment, adverse effects and laboratory results.**
(XLSX)

**S1 Trial. Study protocol.**
(PDF)

## Acknowledgments

The GALILEI (DoG triAL with Impromune in LEIshmania) trial Group includes: HCV Universidad CEU Cardenal Herrera, Valencia (Óscar Cortadellas), Clínica Veterinaria Dinos, Murcia (Laura Sánchez), Hospital Clínico Veterinario, Facultad de Veterinaria, Universidad Complutense de Madrid, Madrid (Guadalupe Miró, Ana Montoya and Juliana Sarquis), Centro Veterinario San Francisco de Asis, Alicante (Manuel Manchón), HCV Universitat Autònoma de Barcelona, Barcelona (Lluís Ferrer and Laura Ordeix), Faculty of Veterinary Medicine, University of Lisbon, Lisbon, Portugal (Rodolfo Oliveira Leal and Maria Joana Dias), Clínica Veterinaria San Jorge, Ibiza (Fernando Ribas), Centro Veterinario Guadiamar, Sevilla (Carmen Acosta), Clínica Veterinaria Díaz de la Cebosa, Guadalajara (Cristina Santos), Hospital Veterinario Althaía, Alicante (Pilar Sagredo), FRAVET-Európolis, Madrid (Cristeta Fraile and Alejandro Ojeda), Hospital Clínico Veterinario UAX, Madrid (Marina Domínguez and Emma Delgado), Clínica Veterinaria Dr. Bernal, Murcia (Luis Bernal) and University of Murcia (Luis Pardo-Marín).

## Author Contributions

**Conceptualization:** Guadalupe Miró, Sergi Segarra.

**Data curation:** Laia Montell, Ester Costa, Joan Teichenne, Roger Mariné-Casadó.

**Formal analysis:** Joan Teichenne, Roger Mariné-Casadó.

**Funding acquisition:** Sergi Segarra.

**Investigation:** Guadalupe Miró.

**Methodology:** Guadalupe Miró, Sergi Segarra, José Joaquín Cerón, Lluís Ferrer, Laia Solano-Gallego, Joan Teichenne, Xavier Roura.

**Project administration:** Sergi Segarra.

**Resources:** Sergi Segarra, José Joaquín Cerón, Laia Montell, Ester Costa.

**Software:** Guadalupe Miró, Sergi Segarra.

**Supervision:** Guadalupe Miró, Sergi Segarra.

**Validation:** Guadalupe Miró, Sergi Segarra.

**Visualization:** Guadalupe Miró, Sergi Segarra.

**Writing – original draft:** Guadalupe Miró, Sergi Segarra.

**Writing – review & editing:** Guadalupe Miró, Sergi Segarra, José Joaquín Cerón, Lluís Ferrer, Laia Solano-Gallego, Laia Montell, Ester Costa, Joan Teichenne, Roger Mariné-Casadó, Xavier Roura.

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
