## [Decision Letter · Decision Letter 0]

24 Apr 2024

Dear Prof Miró,

Thank you very much for submitting your manuscript "New immunomodulatory treatment protocol for canine leishmaniosis reduces parasitemia and proteinuria" for consideration at PLOS Neglected Tropical Diseases. As with all papers reviewed by the journal, your manuscript was reviewed by members of the editorial board and by several independent reviewers. In light of the reviews (below this email), we would like to invite the resubmission of a significantly-revised version that takes into account the reviewers' comments. 

We cannot make any decision about publication until we have seen the revised manuscript and your response to the reviewers' comments. Your revised manuscript is also likely to be sent to reviewers for further evaluation.

Sincerely,

Camila I. de Oliveira

Academic Editor

Hira Nakhasi

Section Editor

Reviewer's Responses to Questions

**Key Review Criteria Required for Acceptance?**

**Methods**

-Are the objectives of the study clearly articulated with a clear testable hypothesis stated?

-Is the study design appropriate to address the stated objectives?

-Is the population clearly described and appropriate for the hypothesis being tested?

-Is the sample size sufficient to ensure adequate power to address the hypothesis being tested?

-Were correct statistical analysis used to support conclusions?

-Are there concerns about ethical or regulatory requirements being met?

Reviewer #1: -Are the objectives of the study clearly articulated with a clear testable hypothesis stated? yes

-Is the study design appropriate to address the stated objectives?no

-Is the population clearly described and appropriate for the hypothesis being tested?no

-Is the sample size sufficient to ensure adequate power to address the hypothesis being tested?yes

-Were correct statistical analysis used to support conclusions?yes

-Are there concerns about ethical or regulatory requirements being met?no

Reviewer #2: The study meticulously articulates its objectives and presents a well-defined, testable hypothesis, ensuring the study design is suitably tailored to achieve the stated goals. The chosen population aligns perfectly with the hypothesis, the sample size is aptly calculated to provide robust power, appropriate statistical methods bolster the conclusions, and all ethical and regulatory standards are scrupulously upheld.

Reviewer #3: The authors were very careful regarding the methodology. There is only one issue regarding the analysis of parasite load using blood, but the authors duly justified it.

**Results**

-Does the analysis presented match the analysis plan?

-Are the results clearly and completely presented?

-Are the figures (Tables, Images) of sufficient quality for clarity?

Reviewer #1: -Does the analysis presented match the analysis plan?no

-Are the results clearly and completely presented?no

-Are the figures (Tables, Images) of sufficient quality for clarity?no

Reviewer #2: The analysis aligns seamlessly with the planned approach, and the results are articulated with commendable clarity and completeness. The visual aids, including tables and images, are of high quality, ensuring clear communication of data. However, a detailed discussion is warranted regarding the observed pattern in the IRIS stages within the supplement group, particularly concerning the potential indication of patient relapses over time.

Reviewer #3: Only in the creatinine graph could the authors draw lines indicating the minimum and maximum reference values, making the figure clearer for the reader.

Discussion

Line 273: the authors need to make it clear in this part of the text that the clinical stability of dogs for another year after interrupting the treatment protocol with MGA and Allopurinol occurs in this region of the European continent. This detail is important because in tropical regions the infection pressure is different due to climatic conditions, among other factors.

**Conclusions**

-Are the conclusions supported by the data presented?

-Are the limitations of analysis clearly described?

-Do the authors discuss how these data can be helpful to advance our understanding of the topic under study?

-Is public health relevance addressed?

Reviewer #1: -Are the conclusions supported by the data presented?yes

-Are the limitations of analysis clearly described?yes

-Do the authors discuss how these data can be helpful to advance our understanding of the topic under study?no

-Is public health relevance addressed?yes

Reviewer #2: The conclusions are well-grounded in the data provided, with a transparent acknowledgment of the analysis's limitations. The authors effectively discuss the data's contribution to advancing our understanding of the subject, also addressing its public health significance. However, an in-depth exploration of the IRIS stage patterns in the supplement group would enrich the discussion, particularly in relation to the potential for patient relapses over time.

Reviewer #3: The conclusions are in accordance with the results obtained and presented in the manuscript.

**Editorial and Data Presentation Modifications?**

Reviewer #1: (No Response)

Reviewer #2: - Line 72: The term "me+2thylglucamine antimoniate" should be corrected to "N-methylglucamine antimoniate."

- The acronym AHCC is not explained in the text; could you please provide its definition?

- Clarification is needed on whether sample analysis was centralized in one lab or conducted separately at the 13 veterinary centers.

- In Figure 4, the meaning of the units U.A. for the total clinical score is unclear. Please clarify this in the axis or figure caption.

Reviewer #3: (No Response)

**Summary and General Comments**

Reviewer #1: (No Response)

Reviewer #2: The treatment combining MGA, Allopurinol, and supplements offers a promising alternative for managing canine leishmaniasis. The authors effectively detail how this regimen improves clinical and laboratory outcomes, emphasizing its oral administration convenience for pet owners.

Reviewer #3: The authors recognize the limitations of the study, especially in relation to possible concomitant diseases in animals, which could have some impact on the results. However, any and all field studies in an endemic area are prone to factors of this nature. However, this does not reduce the relevance of the study, which was very well conducted.

PLOS authors have the option to publish the peer review history of their article (what does this mean?). If published, this will include your full peer review and any attached files.

Reviewer #1: No

Reviewer #2: No

Reviewer #3: Yes: Kelvinson Fernandes Viana
---

## [Editor Report · Decision Letter 1]

6 Aug 2024

Dear Prof Miró,

Thank you very much for submitting your manuscript "New immunomodulatory treatment protocol for canine leishmaniosis reduces parasitemia and proteinuria" for consideration at PLOS Neglected Tropical Diseases. As with all papers reviewed by the journal, your manuscript was reviewed by members of the editorial board and by several independent reviewers. In light of the reviews (below this email), we would like to invite the resubmission of a significantly-revised version that takes into account the reviewers' comments. 

We cannot make any decision about publication until we have seen the revised manuscript and your response to the reviewers' comments. Your revised manuscript is also likely to be sent to reviewers for further evaluation.

Sincerely,

Camila I. de Oliveira

Academic Editor

Hira Nakhasi

Section Editor
---

## [Decision Letter · Decision Letter 2]

27 Oct 2024

Dear Prof Miró,

Thank you very much for submitting your manuscript "New immunomodulatory treatment protocol for canine leishmaniosis reduces parasitemia and proteinuria" for consideration at PLOS Neglected Tropical Diseases. As with all papers reviewed by the journal, your manuscript was reviewed by members of the editorial board and by several independent reviewers. The reviewers appreciated the attention to an important topic. Based on the reviews, we are likely to accept this manuscript for publication, providing that you modify the manuscript according to the review recommendations. 

Sincerely,

Camila I. de Oliveira

Academic Editor

Hira Nakhasi

Section Editor

Reviewer's Responses to Questions

**Key Review Criteria Required for Acceptance?**

**Methods**

-Are the objectives of the study clearly articulated with a clear testable hypothesis stated?

-Is the study design appropriate to address the stated objectives?

-Is the population clearly described and appropriate for the hypothesis being tested?

-Is the sample size sufficient to ensure adequate power to address the hypothesis being tested?

-Were correct statistical analysis used to support conclusions?

-Are there concerns about ethical or regulatory requirements being met?

Reviewer #3: -Are the objectives of the study clearly articulated with a clear testable hypothesis stated? Yes

-Is the study design appropriate to address the stated objectives? Yes

-Is the population clearly described and appropriate for the hypothesis being tested? Yes

-Is the sample size sufficient to ensure adequate power to address the hypothesis being tested? Yes

-Were correct statistical analysis used to support conclusions? Yes

-Are there concerns about ethical or regulatory requirements being met? Yes

Reviewer #4: 1- The objectives of the study clearly articulates with a testable hypothesis

2-The study design is appropriate to address the objectives

3- The population of the study is clearly described

4- The authors had a considerable lost in the number of animals, but in this kind of study is difficult to control this aspect. 

5 Statistical analysis needs some clarification in relation to significance of results, since some parameters seem not to be significant and even in the characteristics of the population, there are some significant differences

**Results**

-Does the analysis presented match the analysis plan?

-Are the results clearly and completely presented?

-Are the figures (Tables, Images) of sufficient quality for clarity?

Reviewer #3: Does the analysis presented match the analysis plan? Yes

-Are the results clearly and completely presented? Yes

-Are the figures (Tables, Images) of sufficient quality for clarity? Yes

Reviewer #4: 1-Yes

2-Yes

3-needs to be more clear, some of them.

**Conclusions**

-Are the conclusions supported by the data presented?

-Are the limitations of analysis clearly described?

-Do the authors discuss how these data can be helpful to advance our understanding of the topic under study?

-Is public health relevance addressed?

Reviewer #3: Are the conclusions supported by the data presented? Yes

-Are the limitations of analysis clearly described? Yes

-Do the authors discuss how these data can be helpful to advance our understanding of the topic under study? Yes

-Is public health relevance addressed? Yes

Reviewer #4: Yes

Yes

Yes

Yes

**Editorial and Data Presentation Modifications?**

Reviewer #3: The authors made changes proposed by the reviewers, aiming to improve the presentation of the manuscript.

Reviewer #4: The manuscript is interesting and shows the possibility to improve the treatment of Canine Visceral Leishmaniasis using supplements. Although it is not possible to define what is AHCC, it would be important to mention in since the abstract that is a supplement used in humans to improve the immune response. Another point, it would be discuss the possible mechanism why these compounds contribute to a better results in the outcome of the infection. Is it related to supression of the inflammatory responses? Increase the immune repsonses associated to a Th1 response?

etc

Lines 187-189 Neither were there differences (control vs supplement group,respectively) in LeishVet

 clinical stage (2.19±0.40 vs 2.37±0.49; p< 0.01) or total clinical score (9.00±5.58 vs 10.11±4.62; p< 0.01). 

There are differences between the two groups concerning clinical stage and total clinical score

Lines 225-226 A significant decrease over time (p= 0.004) in parasite load (blood qPCR) was seen only in the

supplement group (Figure 5). It would be important to perform area under curve, because apparently, the difference is very low.

Lines 232- 233We observe the same issue in the Figure 6 concerning the levels of creatinine, when the authors state a significant difference between the control and supplement groups that seems to occur in the first point before the end of treatment (30 days)

**Summary and General Comments**

Reviewer #3: The study has a good experimental design and responds to what was initially proposed, bringing a new adjuvant therapeutic perspective in the treatment of canine visceral leishmaniasis.

Reviewer #4: The manuscript is interesting and shows the possibility to improve the treatment of Canine Visceral Leishmaniasis using supplements. Although it is not possible to define what is AHCC, it would be important to mention in since the abstract that is a supplement used in humans to improve the immune response. Another point, it would be discuss the possible mechanism why these compounds contribute to a better results in the outcome of the infection. Is it related to supression of the inflammatory responses? Increase the immune repsonses associated to a Th1 response?

etc

Lines 187-189 Neither were there differences (control vs supplement group,respectively) in LeishVet

 clinical stage (2.19±0.40 vs 2.37±0.49; p< 0.01) or total clinical score (9.00±5.58 vs 10.11±4.62; p< 0.01). 

There are differences between the two groups concerning clinical stage and total clinical score

Lines 225-226 A significant decrease over time (p= 0.004) in parasite load (blood qPCR) was seen only in the

supplement group (Figure 5). It would be important to perform area under curve, because apparently, the difference is very low.

Lines 232- 233We observe the same issue in the Figure 6 concerning the levels of creatinine, when the authors state a significant difference between the control and supplement groups that seems to occur in the first point before the end of treatment (30 days)

PLOS authors have the option to publish the peer review history of their article (what does this mean?). If published, this will include your full peer review and any attached files.

Reviewer #3: No

Reviewer #4: No

Figure Files:

Data Requirements:

Reproducibility:

References

---

## [Editor Report · Decision Letter 3]

19 Nov 2024

Dear Prof Miró,

We are pleased to inform you that your manuscript 'New immunomodulatory treatment protocol for canine leishmaniosis reduces parasitemia and proteinuria' has been provisionally accepted for publication in PLOS Neglected Tropical Diseases.

Best regards,

Camila I. de Oliveira

Academic Editor

Hira Nakhasi

Section Editor

Shaden Kamhawi

co-Editor-in-Chief

Paul Brindley

co-Editor-in-Chief

Dear Dr Miró,

Thank you for submitting the revised version of your manuscript. The points raised by the referees have been addressed and the manuscript shall be of interest to the PLoS NTDs community. Most of all, thank you for your continued patience with the review process and apologies for the delays.

---

## [Editor Report · Acceptance letter]

5 Dec 2024

Dear Prof Miró,

We are delighted to inform you that your manuscript, "New immunomodulatory treatment protocol for canine leishmaniosis reduces parasitemia and proteinuria," has been formally accepted for publication in PLOS Neglected Tropical Diseases.

Best regards,

Shaden Kamhawi

co-Editor-in-Chief

Paul Brindley

co-Editor-in-Chief
